# Differences in phenotype between long-lived memory B cells against *Plasmodium falciparum* merozoite antigens and variant surface antigens

**Raphael A. Reyes**[1], **Louise Turner**[2], **Isaac Ssewanyana**[3], **Prasanna Jagannathan**[4,5], **Margaret E. Feeney**[6,7], **Thomas Lavstsen**[2], **Bryan Greenhouse**[7], **Sebastiaan Bol**[1], **Evelien M. Bunnik**[1] *

**1** Department of Microbiology, Immunology & Molecular Genetics, Long School of Medicine, The University of Texas Health Science Center at San Antonio, San Antonio, Texas, United States of America, **2** Centre for translational Medicine & Parasitology, Department of Immunology and Microbiology, University of Copenhagen, and Department of Infectious Diseases, Righospitalet, Copenhagen, Denmark, **3** Infectious Disease Research Collaboration, Kampala, Uganda, **4** Department of Medicine, Division of Infectious Diseases, Stanford University, Stanford, California, United States of America, **5** Department of Microbiology & Immunology, Stanford University, Stanford, California, United States of America, **6** Department of Pediatrics, University of California San Francisco, San Francisco, California, United States of America, **7** Department of Medicine, University of California San Francisco, San Francisco, California, United States of America

* bunnik@uthscsa.edu

**Data Availability Statement:** The authors confirm that all data underlying the findings are fully available without restriction. All relevant data are

## Abstract

*Plasmodium falciparum* infections elicit strong humoral immune responses to two main groups of antigens expressed by blood-stage parasites: merozoite antigens that are involved in the erythrocyte invasion process and variant surface antigens that mediate endo-thelial sequestration of infected erythrocytes. Long-lived B cells against both antigen classes can be detected in the circulation for years after exposure, but have not been directly compared. Here, we studied the phenotype of long-lived memory and atypical B cells to merozoite antigens (MSP1 and AMA1) and variant surface antigens (the CIDRα1 domain of PfEMP1) in ten Ugandan adults before and after local reduction of *P. falciparum* transmission. After a median of 1.7 years without *P. falciparum* infections, the percentage of antigen-specific activated B cells declined, but long-lived antigen-specific B cells were still detectable in all individuals. The majority of MSP1/AMA1-specific B cells were CD95$^+$CD11c$^+$ memory B cells, which are primed for rapid differentiation into antibody-secreting cells, and FcRL5$^-$T-bet$^-$ atypical B cells. On the other hand, most CIDRα1-specific B cells were CD95$^-$CD11c$^-$ memory B cells. CIDRα1-specific B cells were also enriched among a subset of atypical B cells that seem poised for antigen presentation. These results point to differences in how these antigens are recognized or processed by the immune system and how *P. falciparum*-specific B cells will respond upon re-infection.

within the paper and its Supporting information files.

**Funding:** This work was supported by the National Institutes of Health (R01 AI153425 to EMB, F31 AI169993 to RAR, TL1 TR002647 to RAR, U19 AI150741 to BG and PJ, and U19 AI089674 that funds the PRISM cohort). The funders had no role in study design, data collection and analysis, decision to publish, or preparation of the manuscript.

## Author summary

Immunity to malaria depends in part on memory B cells that rapidly start producing antibodies upon the next infection with the malaria-causing parasite *Plasmodium falciparum*. Studying memory B cells, in particular long-lived memory B cells, in people living in malaria-endemic regions is difficult, because every *P. falciparum* infection leads to the activation of existing memory B cells and the formation of new memory B cells. To overcome this problem, we used samples collected from the same people at two time points: (1) during high *P. falciparum* transmission and (2) following effective mosquito control that resulted in a two-year period without *P. falciparum* infection. Analyzing antigen-specific B cells from these two time points allowed us to evaluate which *P. falciparum*-specific B cell subsets remained present in the absence of new infections and could thus be considered long-lived. We found that long-lived memory B cells against erythrocyte invasion antigens expressed the surface markers CD95 and CD11c, while B cells recognizing parasite antigens involved in vascular adhesion and immune evasion did not. These results suggest that the immune response 'sees' and responds to these antigens differently.

## Introduction

Malaria continues to be an enormous public health problem in sub-Saharan Africa [1]. This potentially fatal disease is caused by parasites of the *Plasmodium* genus, of which *P. falciparum* is responsible for most malaria cases and deaths [1]. The development of an effective vaccine against *P. falciparum* plays an important role in the fight to eradicate malaria. However, a major hurdle to overcome in malaria vaccine development is the quick waning of vaccine-elicited immune responses. Most malaria vaccines and vaccine candidates elicit antibodies that inhibit parasite invasion or development, as well as memory B cells that will be activated upon the next *P. falciparum* antigen encounter. To improve the durability of malaria vaccine-induced immune responses, it is important to define the characteristics of long-lived anti-parasite immunity (here defined as persisting for at least one year in the absence of exposure), for example by studying long-lived B cell memory induced by *P. falciparum* infection. The goal of this study was to compare long-lived memory B cell responses to different parasite antigens acquired as the result of natural infection in individuals living in a malaria-endemic region.

In the human host, *P. falciparum* develops through several life cycle stages, of which the asexual replication cycle within erythrocytes is responsible for pathogenesis. During this replication cycle, a single *P. falciparum* merozoite infects an erythrocyte and over the course of 48 hours, divides into 16–32 daughter cells. These newly formed merozoites then burst out of the infected erythrocyte, each ready to invade a new erythrocyte. People living in malaria-endemic regions are repeatedly infected by *Plasmodium* parasites and, as a result of these repetitive exposures, develop immunoglobulin (Ig) M and IgG responses against asexual blood-stage parasites that protect against disease [2–6]. The main antigenic targets of these antibody responses can be divided into two categories. The first class is expressed by merozoites and is involved in erythrocyte invasion, such as merozoite surface protein 1 (MSP1) and apical membrane antigen 1 (AMA1). The second class of antigens comprises several families of variant surface antigens that are expressed by *P. falciparum* on the surface of the infected erythrocyte. The most studied of these are members of the *P. falciparum* erythrocyte membrane protein 1 (PfEMP1) family that mediate binding to endothelial receptors on the host microvasculature.

Merozoite-specific memory B cells have been detected in the circulation up to 16 years after infection in the absence of antigen exposure [7–10]. Similarly, memory B cells against the PfEMP1 variant VAR2CSA involved in pregnancy-associated malaria can be found in the circulation for many years [11]. However, the phenotype of these long-lived *P. falciparum*-specific memory B cells has not been studied in great detail. Antigen-experienced B cells in the circulation can be divided into two main populations: conventional memory B cells (IgD⁻CD27⁺) and double negative (DN) B cells (IgD⁻CD27⁻), both known to harbor *P. falciparum* merozoite antigen-specific B cells [4,12]. Recent studies have identified subsets of conventional memory B cells that are associated with durable humoral immune responses after influenza and tetanus vaccination, and SARS-CoV-2 infection [13–16]. In particular, the surface protein FcRL5, often expressed in conjunction with the integrin CD11c and the transcription factor T-bet, marks a subset of long-lived class-switched memory B cells that are epigenetically and metabolically poised to differentiate into antibody-secreting cells during recall responses [13]. The same three markers are expressed by a subset of DN B cells called DN2 or atypical B cells that are typically expanded in *P. falciparum*-exposed individuals [17,18]. Atypical B cells also have the capacity to differentiate into antibody-secreting cells and may thus contribute to protection against malaria [19,20]. However, it is unknown whether *P. falciparum*-specific atypical B cells have the same durability as *P. falciparum*-specific memory B cells and if there are differences in the phenotype and longevity of B cells specific for merozoite antigens or variant surface antigens.

In malaria-endemic regions, repetitive *P. falciparum* infections result in boosting of the immune response during every infection. These repeated infections complicate the study of the durability of anti-parasite immune responses, because every antigen exposure results in the activation and differentiation of long-lived memory B cells into antibody-secreting cells, as well as the formation of new memory B cells. To overcome this problem, we used samples that were collected before and after local reduction of *P. falciparum* transmission. Analyzing antigen-specific B cells from these two time points allowed us to evaluate which *P. falciparum*-specific B cell subsets remained present in the absence of new infections and could thus be considered long-lived. Using high parameter spectral flow cytometry, we analyzed the phenotype of memory and atypical B cells that recognize *P. falciparum* merozoite antigens or variant surface antigens. Additionally, we performed an unsupervised clustering analysis of antigen-specific B cells to identify characteristics of the long-lived B cell response that are unique to each class of antigens.

## Results

### Activated B cell subsets decreased in abundance in the absence of *Plasmodium falciparum* exposure

To study *P. falciparum*-specific B cell responses that were maintained in the absence of new infections, we used peripheral blood mononuclear cells from ten *P. falciparum*-exposed Ugandan adults collected during a period of high *P. falciparum* transmission and after effective mosquito control by means of indoor residual spraying (IRS) had reduced parasite prevalence by 80% [21]. Individuals had not tested positive for *P. falciparum* infection for a median of 1.7 years (range, 1.1–2.5 years) before collection of blood at the second time point (Fig 1 and S1 Table). B cells were isolated from peripheral blood mononuclear cells and stained with a comprehensive panel of B cell markers (S2 Table). This panel included antigen tetramers that allowed for the detection of B cells with specificity to two classes of *P. falciparum* antigens: (i) merozoite proteins involved in erythrocyte invasion (merozoite surface protein 1 [MSP1] and apical membrane antigen 1 [AMA1]), and (ii) the cysteine-rich interdomain region α1

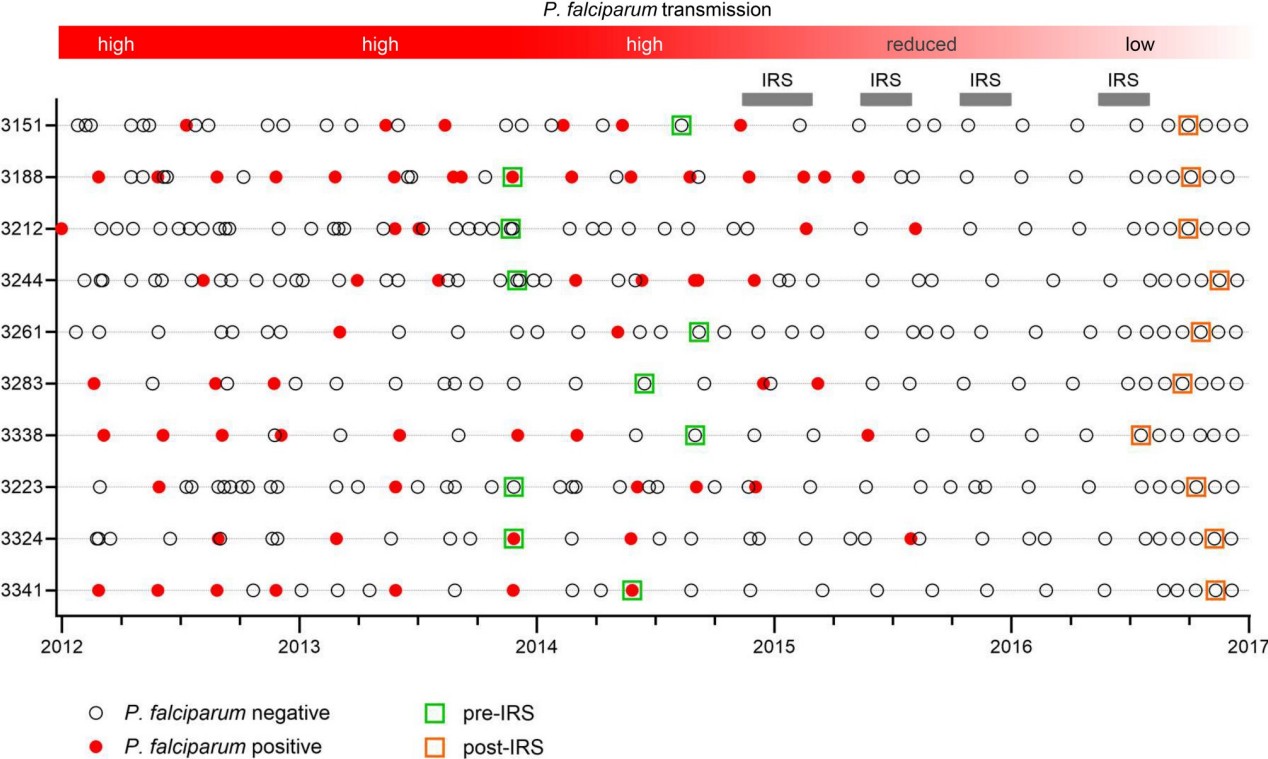

**Fig 1. Timing of sample collection.** Cohort participants (n = 10) were sampled during routine clinic visits roughly every three months and when they visited a study clinic due to illness. For each visit, the outcome of screening for parasitemia is indicated (open circle for negative, closed red circle for positive). Samples used in this study were collected in late 2013 through late 2014 when *P. falciparum* transmission was high (green squares), and during the second half of 2016 after four rounds of indoor residual spraying (IRS) had reduced *P. falciparum* prevalence by 80% (orange squares). The median time between the last known *P. falciparum* infection and collection of the second sample was 1.7 years.

(CIDRα1) domain of the variant surface antigen *P. falciparum* erythrocyte membrane protein 1 (PfEMP1) that is expressed on the surface of the infected erythrocyte. These antigens were selected because we have previously used these proteins to isolate monoclonal antibodies with confirmed antigen specificities [12,22].

To assess the composition of the B cell compartment irrespective of antigen-specificity, we first determined the relative abundance of major B cell populations during high *P. falciparum* transmission and after a reduction in parasite exposure. Following published guidelines for the identification of B cell populations [23], total CD19+CD20+ B cells were first gated on CD24 and CD38 to identify transitional B cells (CD24+CD38+) and plasmablasts (CD24-CD38+). The remaining cells were then divided into naïve B cells (IgD+CD27-), unswitched memory B cells (IgD+CD27+), switched memory B cells (IgD-CD27+), and double negative B cells (IgD-CD27-) (S1 Fig). Based on the expression of CD11c, naïve B cells were further divided into resting (CD11c-) and activated (CD11c+). For unswitched and switched memory B cell populations, we used CD21 to divide cells into resting (CD21+) and activated (CD21-). Using CD11c and CD21, double negative (DN) B cells were separated into sub-populations DN1, DN2, DN3, and DN4 (S1 Fig). DN2 (or atypical; CD11c+CD21-) B cells have been studied extensively in *P. falciparum*-exposed individuals [19,20,24–26], whereas the other three sub-populations of DN B cells have not previously been characterized in the context of malaria. DN1 B cells (CD11c-CD21+) and DN4 (CD11c+CD21+) B cells are thought to be closely related

to resting switched memory B cells, while DN3 B cells (CD11c⁻CD21⁻) may be precursors of atypical B cells [27,28].

Following IRS, we observed a consistent decrease in the percentage of activated cells: naïve B cells from a median of 5.1% to 2.4% (~55% reduction), unswitched memory B cells from 3.4% to 1.3% (~60% reduction), switched memory B cells from 12.3% to 7.6% (~40% reduction) and plasmablasts from 0.2% to 0.1% (~50% reduction) (Fig 2A and S3 Table). In contrast, the percentage of DN1 cells increased significantly post-IRS, from 2.2% to 3.6% (~40% increase) (Fig 2A). These results are in line with a decrease in immune activation due to an interruption of exposure to *P. falciparum* infections. A similar pattern was observed when we analyzed antigen-specific B cells: the relative abundance of activated, but not resting, MSP1/AMA1-specific and CIDRα1-specific naïve B cells, unswitched memory B cells, and switched memory B cells declined between the two time points (~70, 40, and 75% reduction, respectively for MSP1/AMA1-specific B cells and ~85, 75, and 75% reduction, respectively for CIDRα1-specific B cells) (Fig 2B and 2C). Additionally, although the total percentage of atypical B cells did not change, the percentage of MSP1/AMA1-specific and CIDRα1-specific atypical B cells decreased (~60% and 65% reduction, respectively). The percentages of antigen-specific DN1, DN3, and DN4 B cells, as well as plasmablasts were small, with only CIDRα1-specific plasmablasts decreasing over time (Figs 2 and S2). When looking at the abundance of MSP1/AMA1-specific and CIDRα1-specific B cells among total B cells, we observed that both populations decreased in size by about 50%, although these differences were not statistically significant (Fig 2B and 2C).

Collectively, these results suggest that the percentage of *P. falciparum*-specific activated B cells declines in the absence of *P. falciparum* infection but that long-lived *P. falciparum*-specific B cells are detectable in the circulation after more than a year without parasite exposure. In addition, we did not observe a difference in the percentage of B cells with specificity for merozoite antigens or variant surface antigens that were lost.

## Long-lived MSP1/AMA1-specific and CIDRα1-specific B cells differed in phenotype

To further explore similarities and differences in the long-lived memory B cell responses to merozoite antigens and variant surface antigens, we analyzed the phenotype of MSP1/AMA1-specific and CIDRα1-specific B cells. To do this in an unbiased way and without limiting the analysis to pre-defined B cell subsets, we first grouped all non-naïve antigen-specific B cells from all ten individuals at both time points. Unsupervised clustering was then performed based on expression of all markers, except for Ig isotypes to prevent these from dominating the clustering (Figs 3A and S3 and S3 Table). This analysis identified six clusters, which were classified based on the expression of markers associated with previously characterized B cell populations, as follows: plasmablasts (CD27⁺CD38⁺), activated memory B cells (CD21⁻CD27⁺), two atypical B cell subsets (CD21⁻CD27⁻CD11c⁺), and two memory B cell subsets (CD21⁺CD27⁺) (Fig 3B). The two atypical B cell subsets were distinguished by differences in the expression of FcRL5 and T-bet, while the two memory B cell subsets mainly differed in the expression of CD11c and CD95.

Overlay of MSP1/AMA1-specific and CIDRα1-specific B cells onto the composite UMAP of all antigen-specific B cells showed differences in their distribution over the six clusters, especially at the post-IRS time point (Fig 3C). We therefore determined the percentage of antigen-specific cells that belonged to each of the six subsets of B cells (Fig 3D and S3 Table). This allowed us to quantify the changes in composition of the MSP1/AMA1-specific and CIDRα1-specific B cells by calculating the percentage overlap between the samples collected before and

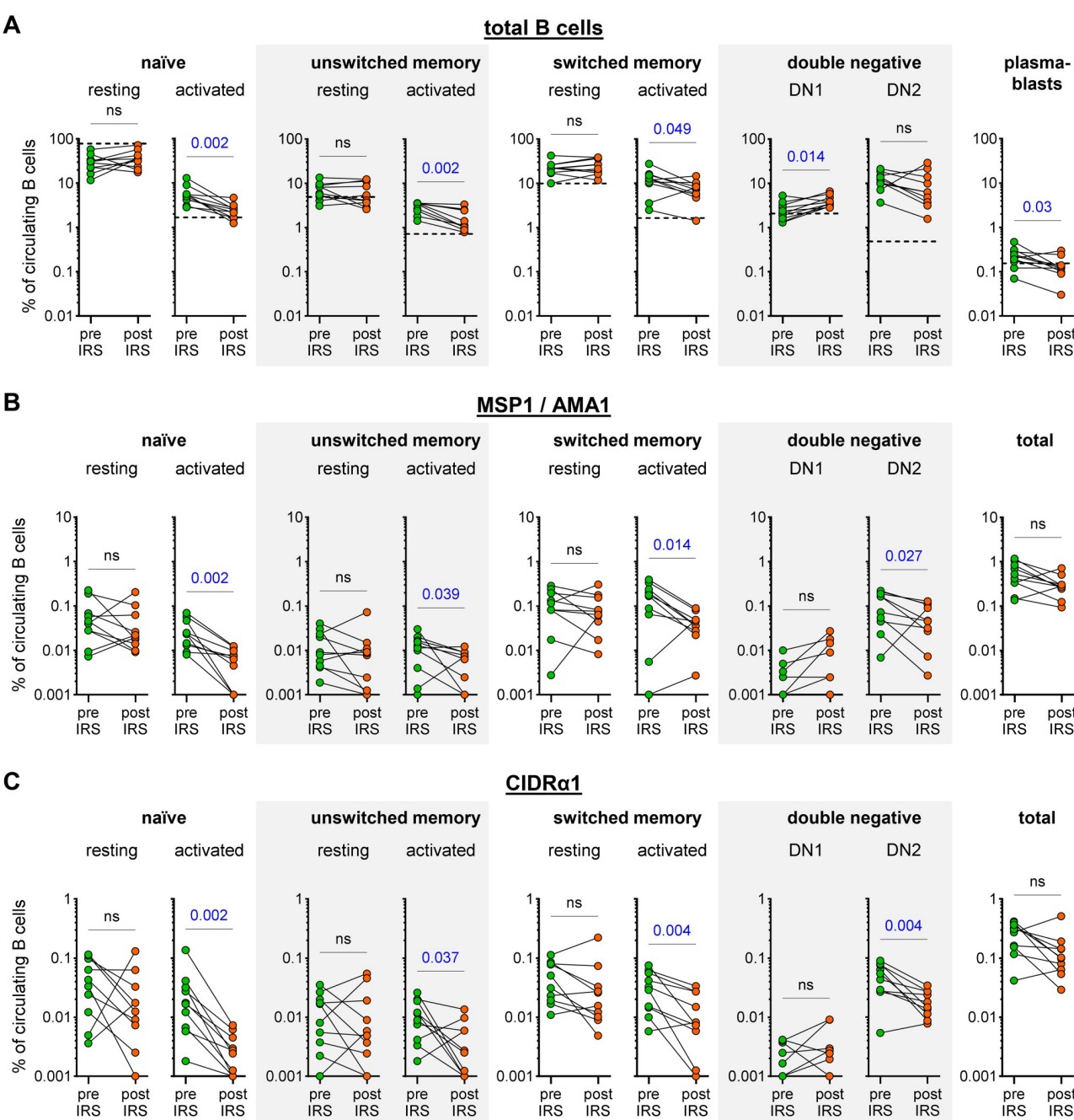

**Fig 2. Abundance of total and antigen-specific B cell subsets in the circulation during high parasite transmission and in the absence of *P. falciparum* exposure.** The percentage of B cell subsets among circulating B cells is shown for total B cells (A), MSP1/AMA1-specific B cells (B), and CIDRα1-specific B cells (C). For MSP1/AMA1-specific B cells and CIDRα1-specific B cells, the total percentage among all circulating B cells is also shown (right most graphs in each panel). All panels show data for all 10 individuals. In panels B and C, no antigen-specific DN1 cells were detected both pre- and post-IRS for four and three individuals, respectively (plotted as 0.001 for visualization purposes, see S3 Table for raw data). These data points therefore overlap and are not clearly visible. Differences between groups were evaluated using a Wilcoxon matched-pairs signed-rank test. P values < 0.05 are shown in blue. As a reference, the median percentages of B cell subsets among circulating B cells in *P. falciparum*-naïve US donors (n = 7) are indicated with dotted lines in panel A. ns, not significant.

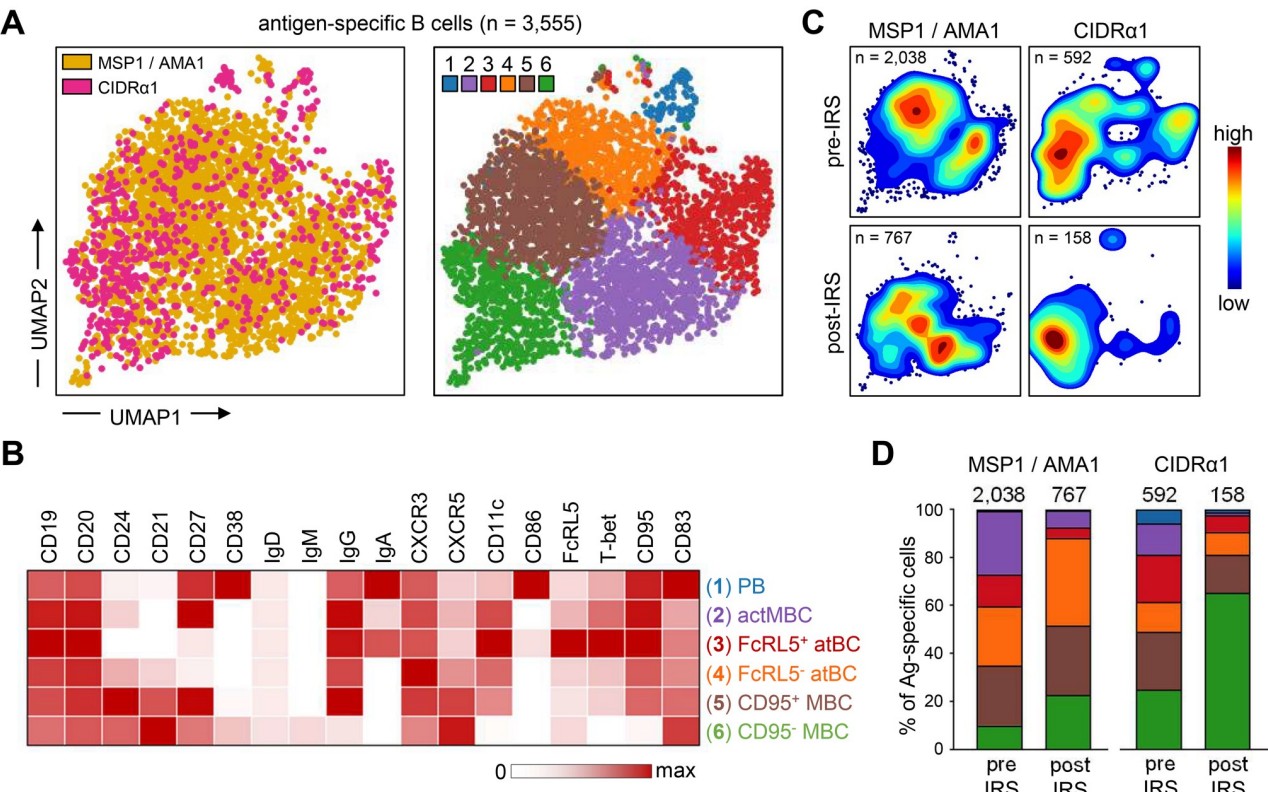

**Fig 3. Differences in phenotype between long-lived MSP1/AMA1-specific and CIDRα1-specific B cells. A)** Composite UMAP of antigen-specific B cells (n = 3,555) from samples collected at two time points from 10 individuals, with B cells colored by antigen-specificity (left) and by cluster (right). **B)** Median fluorescence intensity of 18 surface and intracellular markers in the six B cell subsets, calculated using flow cytometry data from all 20 samples (two time points for 10 individuals) and color-coded relative to the maximum intensity observed among all B cell populations, including naïve B cells not shown in the heatmap (see S3 Table). **C)** Contour plot overlay of MSP1/AMA1-specific and CIDRα1-specific B cells onto the composite UMAP, shown separately for samples collected pre-IRS and post-IRS. **D)** Distribution of all MSP1/AMA1-specific and CIDRα1-specific B cells over the six B cell subsets at the pre-IRS and post-IRS time points. The number above the bar represents the total number of cells. Colors represent the different subsets shown in panel D. PB, plasmablasts; actMBC, activated memory B cells; atBC, atypical B cells; MBC, memory B cells.

after IRS. The overlap in composition of CIDRα1-specific B cells between the two time points (60%) was slightly lower than that of MSP1/AMA1-specific B cells (71%), suggesting that the population of CIDRα1-specific B cells may have undergone more changes during the period with reduced *P. falciparum* transmission. When comparing the two groups of antigen-specific B cells at each time point, we observed that the populations of MSP1/AMA1-specific and CIDRα1-specific B cells were more similar during high parasite transmission (74%) than in the absence of parasite exposure (54%). Thus, the populations of circulating long-lived B cells with specificity for merozoite antigens or variant surface antigens seem to diverge in their composition in the absence of antigen re-exposure.

To better understand these differences in phenotype between long-lived B cells with different antigen-specificities, we looked more closely at the distribution of MSP1/AMA1-specific and CIDRα1-specific B cells over the six subsets. During a time of high parasite exposure, both MSP1/AMA1-specific and CIDRα1-specific B cells were found among all six subsets. With the exception of plasmablasts that were a small population among both antigen groups, antigen-specific B cells were distributed fairly evenly over the various sub-populations (Fig 3D). In line with the reduction in immune activation, the percentage of short-lived plasmablasts (cluster 1) and activated memory B cells (cluster 2) was lower post-IRS (20–70% of the pre-IRS level)

(Fig 3D). The same was true for the FcRL5[+]T-bet[+] subset of atypical B cells (cluster 3), which decreased in proportion by ~65% for both merozoite and variant surface antigens. However, we observed differences in the relative abundance of subsets 4, 5, and 6 between the two antigen groups. While the fraction of FcRL5[-]T-bet[-] atypical B cells (cluster 4) increased among MSP1/AMA1-specific B cells (from 25% pre-IRS to 37% post-IRS), it remained unchanged among CIDRα1-specific B cells (~10%). Additionally, the fraction of CD95[+]CD11c[+] memory B cells (cluster 5) remained stable among MSP1/AMA1-specific B cells (25–30%) but decreased among CIDRα1-specific B cells (from 24% pre-IRS to 16% post-IRS). Together, subsets 4 and 5 made up the majority (65%) of MSP1/AMA1-specific B cells post-IRS, but only 25% of CIDRα1-specific B cells at this time point. In contrast, 65% of all CIDRα1-specific B cells detected post-IRS were CD95[-]CD11c[-] memory B cells (cluster 6), an almost three-fold increase in comparison to its proportion among CIDRα1-specific B cells present during high *P. falciparum* exposure (25%). Collectively, these results suggest that circulating long-lived anti-*Plasmodium* B cells with different antigen specificities may have different phenotypes.

## Long-lived MSP1/AMA1-specific memory B cells expressed CD95 and CD11c

The difference in proportions of CD95[+]CD11c[+] (cluster 5) and CD95[-]CD11c[-] (cluster 6) long-lived memory B cells between merozoite antigen-specific and variant surface antigen-specific memory B cells was interesting given that, in general, the majority of class-switched memory B cells (CD19[+]IgD[-]CD27[+]) do not express these markers [14,16,29]. To study this observation in more detail, we determined the percentages of CD95[+] cells and CD11c[+] cells among antigen-specific switched memory B cells and the total population of switched memory B cells (gated manually as shown in S1A Fig). Irrespective of the level of parasite exposure, the large majority (80–95%) of MSP1/AMA1-specific switched memory B cells expressed CD95, compared to only 40–50% of all switched memory B cells (Fig 4). A similar pattern was seen for CD11c with 70–80% of MSP1/AMA1-specific switched memory B cells expressing CD11c, compared to only 35–40% in the total population of switched memory B cells (Fig 4). During high parasite exposure, the percentages of CD95[+] / CD11c[+] MSP1/AMA1-specific switched memory B cells did not differ from those among CIDRα1-specific switched memory B cells (median, 60–70%), although it should be mentioned that the percentages of CD95[+] / CD11c[+] CIDRα1-specific switched memory B cells were also not significantly different from those of the total population of switched memory B cells (Fig 4). Post-IRS, the percentages of CD95[+] / CD11c[+] CIDRα1-specific switched memory B cells decreased to 25–30% and were significantly different from those among MSP1/AMA1-specific switched memory B cells.

CD11c is often co-expressed with FcRL5 and T-bet. Collectively, these markers have been associated with long-lived class-switched memory B cells that participate in robust recall responses [13,14]. To determine whether MSP1/AMA1-specific memory B cells not only express higher levels of CD11c than CIDRα1-specific switched memory B cells, but also higher levels of these other two markers, we compared the percentages of FcRL5[+] cells and T-bet[+] cells among antigen-specific switched memory B cells and the total population of switched memory B cells. The percentage of T-bet[+] cells among MSP1/AMA1-specific and CIDRα1-specific switched memory B cells was similar during high parasite exposure (median, 30–40%) and post-IRS (median, 5–20%), and was not increased as compared to the total population of switched memory B cells (Fig 4). The percentage of cells expressing FcRL5 was low and did not differ between MSP1/AMA1-specific and CIDRα1-specific switched memory B cells (Fig 4). Thus, CD95 and CD11c, but not other markers of durable immunity, seem to be

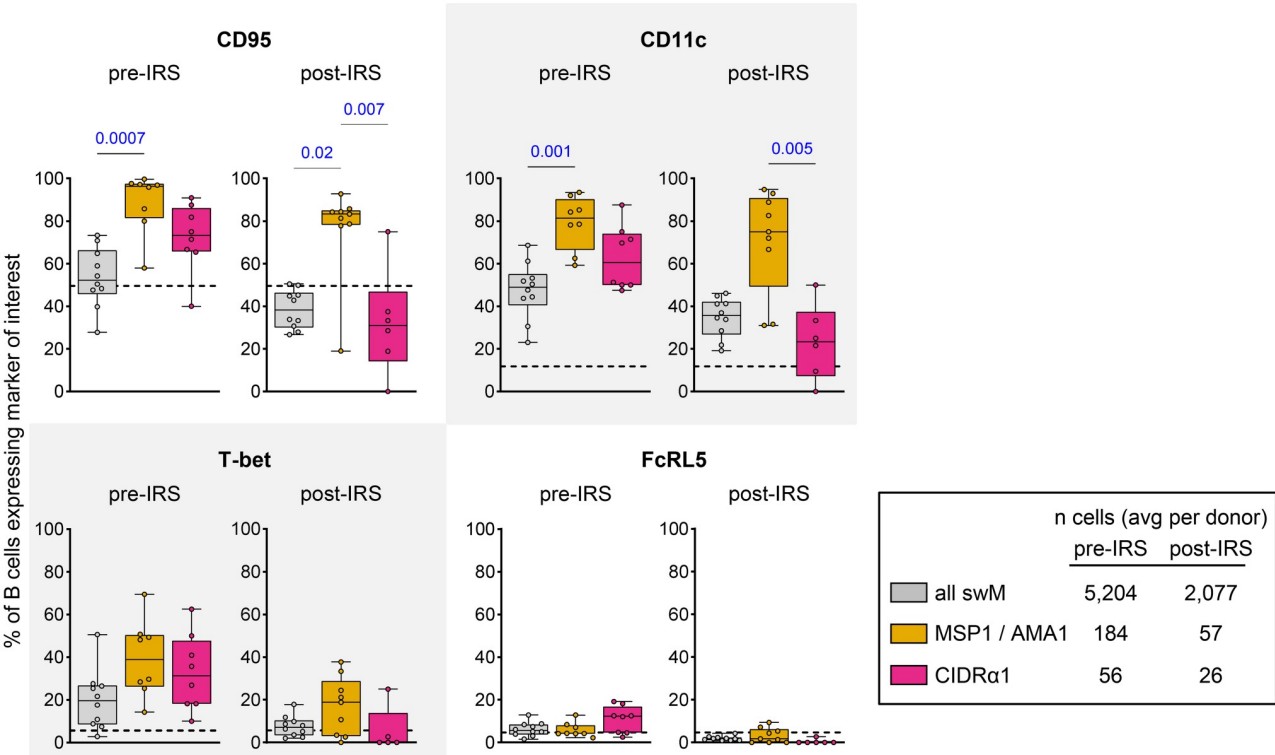

**Fig 4. Expression of CD95, CD11c, T-bet, and FcRL5 among long-lived switched memory B cells.** The percentage of CD95+, CD11c+, T-bet+, and FcRL5+ cells are shown pre-IRS and post-IRS for all switched memory B cells, as well as MSP1/AMA1-specific and CIDRα1-specific switched memory B cells. Differences between groups were tested using a Kruskal-Wallis test, followed by comparisons between all pairs of groups using Dunn's post-hoc test, which reports P values that have been corrected for multiple comparisons. Only P values < 0.05 are shown. As a reference, the median percentages of CD95+, CD11c+, T-bet+, and FcRL5+ switched memory B cells of *P. falciparum*-naïve US donors (n = 7) are indicated with dotted lines. swM, switched memory B cell.

differentially expressed between MSP1/AMA1-specific and CIDRα1-specific long-lived memory B cells present post-IRS.

## CD86+CD11chi atypical B cells were enriched for CIDRα1-specific cells

As shown in Fig 3E, we observed that the large majority (90%) of long-lived MSP1/AMA1-specific atypical B cells (clusters 3 and 4, post-IRS) did not express FcRL5 and T-bet, whereas CIDRα1-specific atypical B cells were divided almost equally between the FcRL5+T-bet+ (cluster 3) and FcRL5-T-bet- (cluster 4) populations. We recently described three subsets of atypical B cells with different functional profiles [20], that we will here refer to as atBC1 (CD86+CD11chi), atBC2 (CD86-CD11chi), and atBC3 (CD86-CD11cint). Of these, atBC1 and atBC2 expressed high levels of FcRL5 and T-bet, whereas atBC3 expressed low levels of T-bet and did not express FcRL5. We determined the proportion of these three previously described subsets of atypical B cells among the MSP1/AMA1-specific atypical B cells and the CIDRα1-specific atypical B cells. After gating on IgD- atypical B cells (CD21-CD27-CD11c+IgD-), unsupervised clustering was performed, followed by separating the cells into the three atypical B cell subsets based on expression of CD11c and CD86 (Figs 5A and S4). Overall, the fraction of atBC1 among all atypical B cells was smallest (25%) and decreased in the absence of *P. falciparum*-exposure (6%) (Fig 5B). The fraction of atBC2 increased from 27% to 40%, while the relative size of subset atBC3 remained stable at ~50%. However, it is important to note that

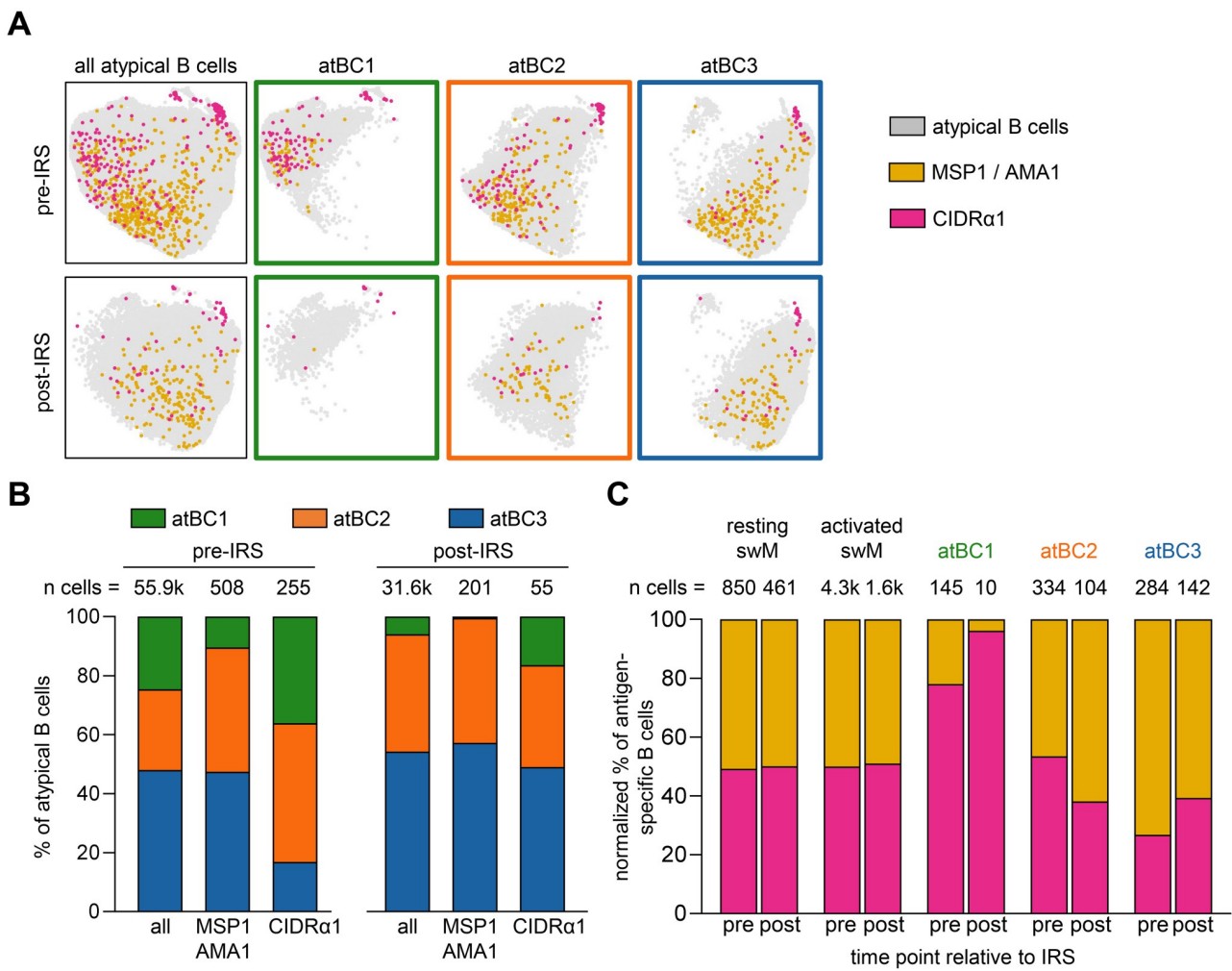

**Fig 5. Distribution of MSP1/AMA1-specific and CIDRα1-specific B cells among atypical B cell subsets. A)** Composite UMAP of atypical B cells from samples collected at two time points from 10 individuals, separated by time point (left column). UMAPs for each of the three atypical B cell subsets are also shown (right three columns in green, orange, and blue). MSP1/AMA1-specific and CIDRα1-specific B cells were projected onto these UMAPs. **B)** Distribution of MSP1/AMA1-specific and CIDRα1-specific B cells over the three atypical B cell subsets in samples collected during times of frequent *P. falciparum* infection (pre-IRS) and in absence of parasite infection (post-IRS). **C)** The normalized percentage of MSP1/AMA1-specific and CIDRα1-specific B cells among different B cell subsets. swM, switched memory B cell; atBC*i*, atypical B cell subset *i*.

atypical B cells of all subsets were still detected in the circulation post-IRS, indicating that these cells are long-lived in the absence of infection.

Next, we determined the proportion of MSP1/AMA1-specific and CIDRα1-specific cells among the three atypical B cell subsets, as well as among resting and activated memory B cells. Antigen-specific B cells were found among all atypical and memory B cell populations, but their distributions within each population differed. Resting memory B cells, activated memory B cells, and atBC2 harbored equal proportions of MSP1/AMA1-specific and CIDRα1-specific cells (Fig 5C), suggesting that there is no enrichment for antigen-specificity among these B cell populations. AtBC3 contained about 60% MSP1/AMA1-specific cells and 40% CIDRα1-specific cells, although this difference did not reach statistical significance when analyzed for individual donors (S5 Fig). In stark contrast, the majority of antigen-specific atBC1 (~80% after normalizing for the total abundance of MSP1/AMA1-specific and CIDRα1-specific cells) was

specific for CIDRα1 at the pre-IRS time point (Figs 5C and S5). The relative abundance of CIDRα1-specific cells within atBC1 was even higher post-IRS, although it should be noted that very few antigen-specific B cells were detected at this time point (10 cells in total). Collectively, these results suggest that atypical B cells are part of the long-lived B cell response to *Plasmodium* antigens and point to differences in the atypical B cell response to different classes of antigens.

## Discussion

Our current understanding of the longevity of naturally acquired immunity against *P. falciparum* is mostly based on studies that measured the presence of anti-parasite antibodies in serum. Previous research on memory B cell responses in *P. falciparum*-exposed individuals has mainly focused on changes in the major B cell populations shortly after acute infection or on studying B cell memory in non-immune individuals [7,30,31]. In recent years, novel subsets of memory and atypical B cells have been discovered, each of which may play a different role in the immune response to infection [13–16,20]. To identify components of the naturally acquired B cell response that contribute to long-lived memory against *P. falciparum* infection, we studied samples collected from individuals living in a malaria endemic region before and after a reduction in *P. falciparum* exposure due to highly effective vector control. Using spectral flow cytometry, we determined the phenotype of B cells with specificity to *P. falciparum* merozoite antigens (MSP1/AMA1) and variant surface antigens (the CIDRα1 domain of PfEMP1). While both groups of antigen-specific B cells showed similar dynamics over time (i.e., a reduction in the percentage of activated B cells), we observed several differences in the composition of circulating long-lived MSP1/AMA1-specific B cells and CIDRα1-specific B cells.

First, we found that the majority of MSP1/AMA1-specific switched memory B cells expressed CD95 and CD11c. CD95 is a death receptor that can induce apoptosis of the cell when it gets bound by its ligand. CD95 is upregulated in germinal center B cells and activated B cells as a mechanism to regulate the humoral immune response and limit inflammation [32]. Seemingly contradictory, it can also play a role in cell survival and proliferation [33]. Until now, it was unclear whether CD95[+] B cells were recent germinal center emigrants that would disappear shortly after the resolution of infection, or whether they were part of a durable memory B cell response. Our observation that approximately 80% of MSP1/AMA1-specific switched memory B cells expressed CD95 after almost two years without *P. falciparum* exposure suggests that these cells are long-lived in the circulation. Glass *et al.* showed that CD95[+] memory B cells were phenotypically more closely related to plasma cells than other memory B cell subsets, suggesting that these cells readily differentiate into antibody-secreting cells upon antigen encounter [29]. Indeed, CD95[+] memory B cells strongly responded to BCR crosslinking, resulting in the phosphorylation of proteins that are part of the BCR signaling cascade [29]. CD11c expression has also been shown to be upregulated by BCR stimulation, and, similar to CD95[+] memory B cells, CD11c[+] B cells are more prone to differentiate into antibody-secreting cells than CD11c[-] B cells [34]. Together, this suggests that MSP1/AMA1-specific switched memory B cells are enriched for an effector memory population that will rapidly differentiate into antibody-secreting cells upon antigen encounter. Interestingly, long-lived MSP1/AMA1-specific switched memory B cells did not express FcRL5 or T-bet, markers that are often co-expressed with CD11c and have been associated with durable B cell immunity and rapid recall responses following vaccination [13,14]. The lack of FcRL5 and T-bet expression in CD11c[+] memory B cells observed here may point to differences in the immune response to vaccination and infection.

The second major difference between MSP1/AMA1-specific B cells and CIDRα1-specific B cells was that the large majority of atypical B cells subset 1 were CIDRα1-specific. We previously reported that this subset of atypical B cells had the most "atypical" phenotype, with highest expression of CD19, CD11c, FcRL5, and T-bet, and upregulated many genes that are involved in antigen presentation and interaction with T cells [20]. This subset of atypical B cells is most likely to home to the spleen where they function as a memory B cell subset, possibly to rapidly respond to systemic infections [35,36]. Their presence in the circulation may be the result of spillover from the spleen, either due to limited space to harbor all newly formed atypical B cells or to distortion of the spleen architecture during *P. falciparum* infections [37]. In the circulation, the proportion of this atypical B cell subset decreased after a period without *P. falciparum* infection. However, it is possible that these cells are longer-lived in the spleen and are more abundant than can be assessed using PBMCs.

What may underlie the differences in phenotype between MSP1/AMA1-specific and CIDRα1-specific memory B cells? One possible explanation may be a difference in the type of cell that initially captures the antigen for presentation to CD4[+] T cells. MSP1 and AMA1 are both shed from the merozoite surface during invasion and are thus present in the circulation in soluble form [38]. Dendritic cells are highly efficient at taking up and presenting soluble antigen [39]. In contrast, PfEMP1 is expressed on the surface of infected erythrocytes, which can be considered 'particles'. Dendritic cells have been shown to be dispensable for mounting an immune response to particulate antigens [40]. Instead, B cells are the primary antigen-presenting cells that take up particulate antigens, including *Plasmodium*-infected erythrocytes [39–41]. Gao *et al.* recently showed that atypical B cells were better at presenting antigen and activating CD4[+] T cells than other B cell subsets [42]. In line with these observations, we show that atypical B cell subset 1 (that seems primed for antigen presentation) was enriched for particle-associated CIDRα1-specific B cells over B cells with specificity for soluble antigens MSP1 and AMA1, suggesting that capturing infected erythrocytes and presenting parasite antigens to T cells may be an important function of this atypical B cell subset. We have previously determined that this subset of atypical B cells can also be stimulated to differentiate into antibody-secreting cells *in vitro* [20], indicating that it may have dual roles in the immune response to *P. falciparum*.

The difference between soluble and membrane-bound antigens may also have a direct effect on how these antigens are perceived by B cells. Atypical B cells have been shown to be restricted to recognition of membrane-bound antigens [43]. The interaction of a B cell with membrane-associated antigen allows the formation of an immunological synapse. Inhibitory receptors expressed by atypical B cells are excluded from this synapse, resulting in B cell receptor signaling and differentiation towards antibody-secreting cells [43]. This could explain why atypical B cell subset 1 that expresses the highest levels of the inhibitory receptor FcRL5 is enriched for recognition of the CIDRα1 domain of membrane-bound protein PfEMP1. It should however be noted that soluble antigen can also be presented effectively in membrane-context by conventional dendritic cells, follicular dendritic cells, and subcapsular macrophages in secondary lymphoid organs, especially when it is part of an immune complex (reviewed in [44]). This would provide a route for atypical B cells to also respond to soluble merozoite antigens, such as MSP1 and AMA1.

Another explanation for the observed differences in phenotype of antigen-specific B cells is that certain B cell receptor properties predispose B cells to a certain fate. Prior studies have shown that while atypical B cells harbor reactivity against *P. falciparum* antigens [4,20], they are also enriched for autoreactivity [45]. Specifically, atypical B cells produce antibodies against the membrane lipid phosphatidylserine, which can induce the destruction of uninfected erythrocytes and contribute to anemia [46]. It has been hypothesized that atypical B cells are derived

from anergic cells with low levels of autoreactivity [47]. During *P. falciparum* infection, these cells could be activated and acquire mutations in the B cell receptor that increase their affinity to *P. falciparum* antigens and at the same time decrease their autoreactivity [47]. In line with this theory, we have previously observed differences in antibody heavy chain V-gene usage between memory and atypical B cells [48]. Specifically, we showed that $V_H$3-48 was overrepresented among IgG$^+$ atypical B cells. Interestingly, we have recently isolated two broadly inhibitory antibodies against CIDRα1 that both used $V_H$3-48 [22]. These observations suggest that intrinsic autoreactivity of their B cell receptor may equip atypical B cells with the potential to recognize *P. falciparum* variant surface antigens.

Finally, it cannot be ruled out that *P. falciparum*-exposed individuals have different lifetime levels and frequencies of exposure to these two groups of antigens that may result in differences in phenotypes of long-lived B cells. With every asexual replication cycle of bloodstage *P. falciparum*, MSP1 and AMA1 are expressed to mediate erythrocyte invasion. B cell responses against these two antigens are thus expected to get boosted with every *P. falciparum* infection. On the other hand, not all PfEMP1 variants contain a CIDRα1 domain [49]. Although the parasite population within an infected individual may collectively express multiple PfEMP1 variants, it is possible that not every infection leads to exposure of the immune system to CIDRα1. However, we observed a similar reduction of activated MSP1/AMA1-specific and CIDRα1-specific cells from the pre-IRS to the post-IRS time point among all major B cell populations: naïve, unswitched memory, switched memory, and atypical B cells. In addition, the unsupervised clustering analysis of antigen-specific B cells identified a population of CIDRα1-specific plasmablasts that were short-lived in the circulation. Together, these observations suggest that the individuals in this study had recently been exposed to CIDRα1 and that a lack of CIDRα1 exposure in itself cannot explain the differences in long-lived memory B cell responses between CIDRα1 and merozoite antigens.

To determine how these different long-lived B cell subsets contribute to protection against *P. falciparum* infection, it would be important to analyze the connection between the cellular repertoire and plasma antibody responses. For *P. falciparum* antigens, a moderate correlation between memory B cell abundance and IgG titers has been observed for some merozoite antigens, but not for others [30,46]. This is in line with studies for other pathogens, that showed a correlation between the percentage of memory B cells and IgG titers for antigens from several viruses and bacteria [50–53], while other studies have reported the absence of such a correlation [53–56]. The lack of a correlation between the magnitude of the memory B cell and the antibody response fits with the prevailing model that memory B cells and plasma cells are two independently controlled arms of the humoral immune system [57,58]. To determine the contribution of different memory B cell subsets to the recall response against *P. falciparum*, it would be interesting to analyze antibody responses upon re-infection. However, none of the individuals included in this study experienced a recorded *P. falciparum* infection post-IRS, preventing us from performing such an analysis.

This study has several limitations. First, the study population is relatively small and homogeneous (ten women between 25 and 65 years of age). Performing a similar analysis in a larger cohort of individuals of different age groups, including children and men, will strengthen our observations. Second, we assessed B cell responses almost 1.5–2 years after the last known *P. falciparum* infection, which is a relatively short time without antigen exposure. Long-lived memory B cells have been detected decades after infection or vaccination [7–11,53]. It would be interesting to survey the ten individuals included in this study again at a later time point. Third, our analyses are restricted to B cell phenotype and did not include functional assessment of the various B cell populations identified here. Finally, we detected relatively few antigen-specific B cells, especially for CIDRα1 in the absence of *P. falciparum* exposure, which in

some cases prevented us from analyzing responses for each person separately. In these instances, we aggregated antigen-specific B cells from ten individuals and analyzed them in bulk. Using antigen probes of additional CIDRα1 variants and non-3D7 *P. falciparum* MSP1/AMA1 variants could facilitate the detection of larger numbers of antigen-specific B cells. All in all, it will be important to perform additional studies of the phenotype and functionality of long-lived B cells with specificity for *P. falciparum* antigens to reproduce and extend our findings.

In conclusion, we analyzed long-lived B cell responses against merozoite antigens and variant surface antigens in individuals living in a malaria-endemic region at a time when *P. falciparum* transmission was high, and after at least a year (median of 1.7 years) without parasite exposure. The loss of MSP1/AMA1-specific and CIDRα1-specific B cells in the circulation was similar, but the phenotype of long-lived MSP1/AMA1-specific and CIDRα1-specific B cells appeared to differ. The majority of long-lived MSP1/AMA1-specific were CD95$^+$CD11c$^+$ memory B cells and FcRL5$^-$T-bet$^-$ atypical B cells, whereas the majority of long-lived CIDRα1-specific B cells were CD95$^-$CD11c$^-$ memory B cells. Our results do not necessarily point to a qualitative difference in the memory B cell response to these antigens but may be reflective of differences in how these different antigens are recognized or processed by the immune system, and how the immune response will unfold during a new *P. falciparum* infection.

## Materials and methods

### Ethics statement

All ten individuals included in this study were residents of the Nagongera sub-county in Tororo District, Uganda. This region was historically characterized by extremely high *P. falciparum* transmission intensity, with an estimated annual entomological inoculation rate of 125 infectious bites per person per year [59]. Since 2015, multiple rounds of indoor residual spraying (IRS) have dramatically reduced malaria incidence compared with pre-IRS levels [21]. Individuals were selected for inclusion into this study based on age and *P. falciparum* exposure. All individuals included in this study were enrolled in The Program for Resistance, Immunology, Surveillance, and Modeling of Malaria (PRISM) cohort [60] and have provided written consent for the use of their samples for research. All infections (*P. falciparum* and other causes) in the three months prior to sample collection are indicated in S3 Table. Additional information about PRISM participants, including medical history, is available through ClinEpiDB (clinepidb.org). The PRISM cohort study was approved by the Makerere University School of Medicine Research and Ethics Committee (SOMREC), London School of Hygiene and Tropical Medicine IRB, the University of California, San Francisco Human Research Protection Program, and the Stanford University School of Medicine IRB. The use of cohort samples for this study was approved by the Institutional Review Board of the University of Texas Health Science Center at San Antonio.

Samples from *P. falciparum*-naïve US donors (n = 7) were received de-identified from the University of Texas Health San Antonio COVID-19 Repository and were obtained from individuals who had not been SARS-CoV-2-infected for at least 20 weeks prior to sample collection. This repository was reviewed and approved by the University of Texas Health Science Center at San Antonio Institutional Review Board. All study participants provided written informed consent prior to specimen collection for the repository to include use of left-over clinical specimens for research. The COVID-19 Repository utilizes an honest broker system to maintain participant confidentiality and release de-identified data or specimens to recipient investigators.

## B cell isolation

Cryopreserved PBMCs were thawed and immediately mixed with pre-warmed (37°C) thawing medium (IMDM/GlutaMAX supplemented with 10% heat-inactivated FBS (USA origin) and 0.01% Universal Nuclease (Thermo, #88700)) to wash away the DMSO. After centrifugation (250 × g, 5 min at RT), the cell pellet was resuspended in warm thawing medium and viable cells were counted. Next, cells were centrifuged (250 × g, 5 min at RT), resuspended in isolation buffer (PBS with 2% FBS and 1 mM (f/c) EDTA) at 50 million live cells/mL, and filtered through a 35 μm sterile filter cap (Corning, # 352235) to break apart any aggregated PBMCs. B cells were isolated by negative selection using the EasySep Human B Cell Isolation Kit (StemCell, #17954) or the MojoSort Human Pan B Cell Isolation Kit (BioLegend, # 480082) according to the manufacturer's instructions.

## Staining for spectral flow analysis

C-terminally biotinylated full-length *P. falciparum* 3D7 MSP1 and AMA1 were produced in Expi293F cells (Thermo, # A14635) as described previously [12]. C-terminally StrepTagII labeled HB3VAR03 CIDRα1.4, ITVAR22 CIDRα1.7, and IT4VAR20 CIDRα1.1 were produced in baculovirus-infected insect cells as described previously [61]. Antigen tetramers were synthesized by incubating protein with fluorophore-conjugated streptavidin overnight at 4°C at a molar ratio of 6:1 with rotation. MSP1 and AMA1 tetramers were made with APC-conjugated streptavidin (Cytek, # 20-4317-U100) and BUV563-conjugated streptavidin (BD, # 612935), while CIDRα1 tetramers were generated with PE-labeled streptavidin (Cytek, # 50-4317-U100) and BUV661-conjugated streptavidin (BD, # 612979). B cells isolated by negative selection were washed with PBS, centrifuged (250 × g, 5 min), resuspended in 1 ml of PBS containing 1 μl live/dead stain (Zombie UV Fixable Viability kit (Biolegend, # 423107)) and incubated on ice for 30 min. Cells were subsequently washed with cold PBS containing 1% BSA (250 × g, 5 min, 4°C), resuspended with a cocktail of 25 μM of each merozoite tetramer (MSP1/AMA1) diluted in PBS containing 1% BSA to a volume of 100 μl, and incubated at 4°C for 30 min. The cells were then washed twice with cold PBS containing 1% BSA (250 × g, 5 min, 4°C) and incubated with a cocktail of 25 μM of each CIDRα1 tetramer (CIDRα1.1/CIDRα1.4/CIDRα1.7) diluted in PBS containing 1% BSA to a volume of 100 μl, and incubated at 4°C for 30 min. Next, the cells were washed twice with cold PBS containing 1% BSA (250 × g, 5 min, 4°C) and incubated at 4°C for 30 min with a B cell surface marker antibody cocktail (S2 Table) with 10 μl Brilliant Stain Buffer Plus (BD, # 566385) diluted in PBS containing 1% BSA to a volume of 100 μl. The cells were then washed with cold PBS containing 1% BSA (250 × g, 5 min, 4°C), resuspended in 1 ml of Transcription Factor Fix/Perm Concentrate (Cytek, part of # TNB-0607-KIT), diluted with 3 parts Transcription Factor Fix/Perm Diluent (Cytek), and incubated at 4°C for 1 hour. After the incubation, the cells were washed twice with 3 ml of 1× Flow Cytometry Perm Buffer (Cytek) (300 × g, 8 min, 4°C) and resuspended in 1× Flow Cytometry Perm Buffer with an anti-human T-bet antibody. After an incubation at 4°C for 30 min, the cells were washed twice with 3 ml cold 1× Flow Cytometry Perm Buffer (300 × g, 8 min, 4°C) and once with 3 ml cold PBS containing 1% BSA, resuspended in cold PBS containing 1% BSA to 20–30 million cells/ml and filtered into a FACS tube through a 35 μm sterile filter cap. Cells were analyzed by flow cytometry immediately following intracellular staining.

## Spectral flow cytometry analysis

B cells were analyzed on a Cytek Aurora spectral flow cytometer equipped with five lasers. SpectroFlo QC Beads (Cytek, # N7-97355) were run prior to each experiment for routine

performance tracking. Daily quality control and Levey-Jennings tracking reports were used to ensure optimal performance of the machine and to confirm that settings between different runs were comparable. Pooled B cells from two malaria-naïve US donors were used for the unstained control, technical replicates, and to perform compensation for the live/dead stain. UltraComp eBeads Plus Compensation Beads (Thermo, #01-3333-41) were used to perform compensation for all other fluorophores. Between experiments performed on different days, the technical replicates showed near perfect correlation between the expression of cell surface and intracellular markers (Spearman r = 0.98). To minimize experimental variation, paired samples were analyzed within the same experiment.

The cytometry analysis software OMIQ (Dotmatics) was used for all data analysis. B cell subsets and antigen specific B cells were manually gated in OMIQ (S1 Fig). Since both MSP1 and AMA1 tetramers were generated in the same two fluorochrome format (APC and BUV563), MSP1/AMA-specific B cells were collectively defined as cells staining positive for both tetramer formats. For CIDRα1, we used the domain variants IT4VAR20 CIDRα1.1, HB3VAR03 CIDRα1.4, and IT4VAR22 CIDRα1.7 that are highly diverse in sequence. Because tetramers for each CIDRα1 variant were generated using PE and BUV661 fluorochromes, B cells binding either one of the two variants were indistinguishable. Therefore, all CIDRα1.1, CIDRα1.4, and CIDRα1.7-reactive B cells were collectively referred to as CIDRα1-specific B cells. Antigen-specific B cells were defined as cells staining positive for both tetramer formats in a single antigen group. To exclude any non-specific binders, B cells with reactivity to both MSP1/AMA1 and CIDRα1 probes were removed. Thus, the gating strategy for antigen-specific B cells is summarized as follows: single / live / CD19$^+$ / CD20$^+$ / IgD- / non-strep / (MSP1/AMA1$^+$ or CIDRα1$^+$).

Data integration and dimension reduction analysis were performed using Uniform Manifold Approximation and Projection (UMAP). UMAPs of antigen-specific B cells were created using the expression of CD19, CD20, CD21, CD24, CD27, CD38, CD83, CD86, CD95, CXCR3, CXCR5, CD11c, FcRL5, and T-bet as features with default parameters (neighbors = 15, minimum distance = 0.8, metric = Euclidean, random seed = 2478) and included all 20 Ugandan donor samples used in this study for initial projection. FlowSOM [62] was used to identify six cell subsets based on the expression of CD19, CD20, CD21, CD24, CD27, CD38, CD83, CD86, CD95, CXCR3, CXCR5, CD11c, FcRL5, and T-bet (metric = Elucidean, random seed = 1150). For the analysis of atypical B cells, atypical B cells were pre-gated on single / live / CD19$^+$ / CD20$^+$ / CD21$^-$ / CD27$^-$ / CD11c$^+$ / IgD$^-$ cells, followed by the generation of a UMAP (neighbors = 15, minimum distance = 0.8, metric = Euclidean, random seed = 2742) based on the expression of markers associated with atypical B cells (CD19, CD20, CD24, CD38, CD86, CD95, CXCR3, CXCR5, CD11c, FcRL5, and T-bet). To define the three atypical B cell subsets, FlowSOM [62] was used to identify three clusters based on CD11c and CD86 expression using default parameters (metric = Euclidean, random seed = 7333). For the projection of antigen-specific B cells onto the UMAP, gates were manually set to identify populations of interest using two-dimensional displays, which were then overlaid onto the UMAP projection. Mean fluorescence intensities of cell surface and intracellular markers in select B cell subsets were calculated using the heatmap function in OMIQ. For the analysis of expression of individual markers (CD95, CD11c, T-bet, and FcRL5) in switched memory B cells, samples with fewer than 10 cells were excluded from analysis.

## Percent overlap in B cell populations

To determine the similarity in distribution of MSP1/AMA1-specific and CIDRα1-specific B cells over six B cell subsets, we calculated the percentage overlap between pair-wise

combinations of samples. For each subset, we determined which of the two samples had the smallest fraction of that subset. For example, if subset A took up 25% of sample x and 17% of sample y, we used the value of 17%. We then calculated the sum of these six smallest fractions to obtain the overlap in composition between the two samples. All distributions and calculations of percent overlap are included in S3 Table.

## Statistical analysis

Statistical analyses of flow cytometry data were performed in GraphPad Prism 10 with details of statistical tests provided in the relevant figure legends. P values < 0.05 were considered statistically significant.

## Supporting information

**S1 Fig. Gating strategy. A)** The gating scheme used to identify resting and activated B cell populations, as well as subpopulations of double negative cells. The table lists all abbreviations of B cell populations and their definitions. **B)** The gating strategy used to identify antigen-specific B cells. Left: cells that bound to both CIDRα1 and MSP1/AMA1 tetramers (denoted as "strep") were considered non-specific binders. Only "non-strep" cells were used to gate on antigen-specific B cells. Middle: background of antigen-specific B cells in a *P. falciparum*-naïve US donor. Right: detection of antigen-specific B cells in a *P. falciparum*-exposed Ugandan donor. **C)** The gating strategy used to identify CD95+, CD11c+, T-bet+, and FcRL5+ B cells.
(TIF)

**S2 Fig. The percentage of DN3 B cells, DN4 B cells, and plasmablasts among circulating B cells.** The percentages are shown for total B cells (left), MSP1/AMA1-specific B cells (middle), and CIDRα1-specific B cells (right). Differences between groups were evaluated using a Wilcoxon matched-pairs signed-rank test.
(TIF)

**S3 Fig. Projection of the surface and intracellular markers used to generate the composite UMAP of antigen-specific B cells onto this UMAP.**
(TIF)

**S4 Fig. CD11c and CD86 expression projected onto the UMAP of all atypical B cells as well as each of the three subsets individually.**
(TIF)

**S5 Fig. The normalized percentage of MSP1/AMA1-specific and CIDRα1-specific B cells among different B cell subsets.** Values were calculated for individual donors. For the three atypical B cell subsets, donors were only included if the total size of the subset allowed for the detection of at least one MSP1/AMA1-specific B cell and one CIDRα1-specific B cell. For example, if the total percentage of MSP1/AMA1-specific B cells is 1.5%, a B cell population needs to contain at least $1 / 1.5 \times 100 = 67$ cells to be included in this analysis. Because atypical B cell subsets in samples obtained post-IRS were too small to perform this analysis, only pre-IRS data is shown. swM, switched memory B cell; atBC*i*, atypical B cell subset *i*. n.s., not significant.
(TIF)

**S1 Table. Information about individuals included in the study.**
(XLSX)

**S2 Table. Flow cytometry panel.**
(XLSX)

**S3 Table. All data underlying the results.**
(XLSX)

## Acknowledgments

Plasmids encoding 3D7 MSP1-bio, AMA1-bio, and BirA, were a kind gift from Dr. Gavin Wright (Wellcome Sanger Institute). Data were generated in the Flow Cytometry Shared Resource at UT Health San Antonio, which is supported by a grant from the National Cancer Institute (P30 CA054174) to the Mays Cancer Center, a grant from the Cancer Prevention and Research Institute of Texas (CPRIT) (RP210126), a grant from the National Institutes of Health (S10 OD030432), and support from the Office of the Vice President for Research at UT Health San Antonio.

## Author Contributions

**Conceptualization:** Raphael A. Reyes, Evelien M. Bunnik.

**Formal analysis:** Raphael A. Reyes, Sebastiaan Bol, Evelien M. Bunnik.

**Funding acquisition:** Evelien M. Bunnik.

**Investigation:** Raphael A. Reyes, Sebastiaan Bol, Evelien M. Bunnik.

**Project administration:** Evelien M. Bunnik.

**Resources:** Louise Turner, Isaac Ssewanyana, Prasanna Jagannathan, Margaret E. Feeney, Thomas Lavstsen, Bryan Greenhouse.

**Supervision:** Evelien M. Bunnik.

**Visualization:** Raphael A. Reyes, Sebastiaan Bol, Evelien M. Bunnik.

**Writing – original draft:** Raphael A. Reyes, Sebastiaan Bol, Evelien M. Bunnik.

**Writing – review & editing:** Raphael A. Reyes, Louise Turner, Isaac Ssewanyana, Prasanna Jagannathan, Thomas Lavstsen, Bryan Greenhouse, Sebastiaan Bol, Evelien M. Bunnik.

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
