## [Decision Letter · Decision Letter 0]

25 Sep 2024

Dear Dr. Bunnik,

Thank you very much for submitting your manuscript "Differences in phenotype between long-lived memory B cells against Plasmodium falciparum merozoite antigens and variant surface antigens" for consideration at PLOS Pathogens. As with all papers reviewed by the journal, your manuscript was reviewed by members of the editorial board and by several independent reviewers. The reviewers appreciated the attention to an important topic. Based on the reviews, we are likely to accept this manuscript for publication, providing that you modify the manuscript according to the review recommendations.

All three reviewers comment on the value of the characterization of parasite antigen-specific memory B cell populations in individuals during periods of high and low P. falciparum transmission. A key strength of the work includes the tetramer-based antigen specific approaches and rigorous interrogation of the phenotype of AMA-, MSP-, and PfEMP1-specific subsets. However, reviewer 1 and 3 ask that clarifications be provided, as well as inclusion of data that quantifies the absolute numbers of memory B cells. Please also address the data display in Figure 2 and conciser discussion of the limitations of focusing only on IgG and not including analyses of IgM.

Sincerely,

Noah S. Butler

Guest Editor

PLOS Pathogens

Margaret Phillips

Section Editor

PLOS Pathogens

Michael Malim

Editor-in-Chief

PLOS Pathogens

orcid.org/0000-0002-7699-2064

Reviewer Comments (if any, and for reference):

Reviewer's Responses to Questions

**Part I - Summary**

Reviewer #1: The manuscript by Reyes RA, et al., provides a phenotypic description of Plasmodium falciparum-specific (MSP1/AMA1 and CIDRAa1) B cells in Ugandan adults during a period of high transmission and a subsequent period of low transmission, with about 1.7 years between the two time points. This study design allowed for a numeric and phenotypic assessment of long-lived antigen-specific B cells. The authors have provided a well-written assessment of their observations, with most conclusions supported by the data. These data provide further insight into the presence and potential contribution of atypical B cells towards anti-Plasmodium immunity, the maintenance and phenotype of Plasmodium-specific B cells in the absence of continuous antigen exposure, and the impact of Plasmodium antigen distribution between merozoite and surface of infected red blood cells on the phenotype of antigen-specific B cells. Collectively, these data will be a valuable resource to the malaria immunology community.

Reviewer #2: As our tools expand, our ability to identify heterogenous populations of cell types increases. In the case of malaria, there is a high interest in determining if atypical MBCs generated as early as the first exposure to the parasite in humans can play a protective role in recall responses against the parasite. Here, the authors wanted to address how circulating memory B cell (MBC) populations against two different classes of antigens (merozoite proteins involved in invasion and variant surface antigen) resemble or differ from one another after a period of high Plasmodium transmission and after the absence of infection for a median of ~1.7 yrs. The novelty here involves the utilization of B cell tetramers specific for MSP1/AMA-1 and the CIDRa1 domain of PFEMP-1. Spectral flow cytometry was used for phenotyping the MBC populations in the blood of ten Ugandan patients who were involved in a P. falciparum transmission reduction program. Importantly, the authors found that the MBCs generated against the different antigens differed after IRS, with the CIDRa1-specific MBCs primarily resembling atypical MBCs. In contrast, the phenotype of the MSP-1/AMA-1–specific MBCs resembled that of cells primed for activation or atypical MBCs. Overall, the authors conclude that different parasite-derived antigens drive the production of different phenotypes of B cells. This is a finding that will be of interest to others in the field.

While this study provides new insight into the ability of different classes of parasite antigens to give rise to MBCs of varying phenotypes, the study was constrained by several limitations, including being able to address whether these different MBC phenotypes result in functional differences upon recall remained unanswered. Also, the authors did not compare the degree of somatic hypermutations accumulated in the BCR of the different antigen-specific MBC populations, as differences may exist. However, the ability to address the latter question could be due to the limited availability of cells from the patients. Nevertheless, the absence of these data detracts from the findings' overall significance, which is largely a phenotypic characterization of MBC populations, as acknowledged by the authors.

Reviewer #3: This manuscript by Evelien Bunnick and colleagues utilizes a longitudinal study to examine the longevity of memory B cells to both merozoite (MSP1 / AMA1) and the CIDR�1 domain of PfEMP-1 in the absence of continued transmission. The strengths of the work are the thorough analysis of the B cell population in human samples using tetramers and comprehensive flow cytometry, not an easy task given that it is difficult to detect antigen-reactive B cell subsets as they are generally much smaller populations, and the discovery of different subsets of memory B cells for merozoite epitopes vs the infected RBC membrane epitope.

The major weakness of the work in my opinion is that there are only 10 patient samples tested raising a potential issue regarding the replicability of the data. Having said that I would be supportive of publishing the data in this study as it adds something to the literature, in my opinion more than previous reviewers have stated.

**Part II – Major Issues: Key Experiments Required for Acceptance**

Reviewer #1: Lines 276-288 – It is not clear the data support the broad conclusion that CD95 and CD11c are differentially expressed between MSP1/AMA1- and CIDRa1-specific long-lived memory B cells as the percentage of these antigen-specific B cells are not different at the pre-IRS time point. That said, these markers are different between these antigen-specific B cells at the post-IRS timepoint. If the conclusion of this data set is that post-IRS represents “long-lived memory B cells” then the claim is supported. If so, then the authors should make it clear that they are referring specifically to the expression of CD95 and CD11c on the post-IRS time point, which they believe represents long-lived memory B cells.

Reviewer #2: (No Response)

Reviewer #3: There are a couple of data presentation issues I think should be addressed that would help the reader. I would consider the most important omission is the number of cells. All conclusions are based on % of total B cells. However, to be prudent I would like to see the absolute number calculated from the number of circulating PBMCs / B cells. If the incidence of malaria transmission indeed has been reduced in the area are the number of PBMCs in each matched patient sample different at the 2 time points? The answer to this question may well be that it is not but I think it would be important to include this information in some way, perhaps in the supplemental information. Accounting for differences in numbers relative to % is important for interpretation purposes.

Similarly I would like to see a slightly different representation of the % data to help with visual interpretation. The axis in Figure 2 are all different. Whilst the authors have likely chosen to represent the data in this way to make differences between the two sampling time points clearer, it is harder for the reader to compare the changes visually between the different B cell populations as they may be represented in the bloodstream.

**Part III – Minor Issues: Editorial and Data Presentation Modifications**

Reviewer #1: (No Response)

Reviewer #2: Several concerns were raised during the initial review of the manuscript; in the resubmission, the authors largely addressed these concerns by adjusting figures and tables, tempering their conclusions, and adding to their discussion.

Reviewer #3: in line 63 I would also mention IgM – IgM has an underappreciated role in protection and there are papers (Pepper /Crompton / Pierce amongst others) on IgM that should be briefly discussed (I understand why isotypes are not a major feature of the analysis in this study but nonetheless humoral immunity to malaria is not all about IgG). IgM memory cells also exist.

In line 68 – PfEMP-1 is certainly the most studied but I am not entirely sure it is necessarily the most important. For clarity maybe it might be wise to mention it is not the only variant family in falciparum.

PLOS authors have the option to publish the peer review history of their article (what does this mean?). If published, this will include your full peer review and any attached files.

Reviewer #1: No

Reviewer #2: No

Reviewer #3: No

Figure Files:

Data Requirements:

Reproducibility:

References:

---

## [Editor Report · Decision Letter 1]

12 Oct 2024

Dear Dr. Bunnik,

We are pleased to inform you that your manuscript 'Differences in phenotype between long-lived memory B cells against Plasmodium falciparum merozoite antigens and variant surface antigens' has been provisionally accepted for publication in PLOS Pathogens.

Best regards,

Noah S. Butler

Guest Editor

PLOS Pathogens

Margaret Phillips

Section Editor

PLOS Pathogens

Michael Malim

Editor-in-Chief

PLOS Pathogens

orcid.org/0000-0002-7699-2064
---

## [Editor Report · Acceptance letter]

22 Oct 2024

Dear Dr. Bunnik,

We are delighted to inform you that your manuscript, "Differences in phenotype between long-lived memory B cells against Plasmodium falciparum merozoite antigens and variant surface antigens," has been formally accepted for publication in PLOS Pathogens.

Best regards,

Michael Malim

Editor-in-Chief

PLOS Pathogens

orcid.org/0000-0002-7699-2064